# Opioid Treatment in Primary Care: Knowledge and Practical Use of Opioid Therapy

**DOI:** 10.3390/healthcare12020217

**Published:** 2024-01-16

**Authors:** Aleksander Michał Biesiada, Aleksandra Ciałkowska-Rysz, Agnieszka Mastalerz-Migas

**Affiliations:** 1Family Physician Office S&M Ltd., 31-123 Krakow, Poland; 2Polish Society of Family Medicine, 51-141 Wroclaw, Poland; 3Department of Palliative Medicine, Faculty of Medicine, Medical University of Lodz, 90-647 Lodz, Poland; 4Department of Family Medicine, Piast of Silesia Medical University, 50-367 Wroclaw, Poland; agnieszka.mastalerz-migas@umed.wroc.pl

**Keywords:** pain management, primary health care, general practitioners, analgesics, opioid

## Abstract

Background: Primary care physicians play a key role in initiating opioid therapy. However, knowledge gaps in opioid use and pain management are significant barriers to providing optimal care. This research study aims to investigate the educational needs of primary care physicians regarding opioid therapy and opioid use in pain management. Methods: A computer-assisted web interview (CAWI) protocol was used to collect data from primary care physicians. Drug selection criteria, knowledge of opioid substitutes and dosage, and practical use of opioid therapy were evaluated. Results: While 84% of participating physicians (724 respondents) reported initiating opioid treatment, only a minority demonstrated accurate opioid dosage calculations. Significant discrepancies between physicians’ self-perceived knowledge and their clinical skills in opioid prescribing and pain management were observed. In total, 41% of physicians incorrectly indicated dose conversion rates for tramadol (the most frequently used drug according to 65% of responders). Conclusions: Targeted educational programs are essential to bridge the knowledge gap and increase physicians’ competence in pain management. The proper self-assessment of one’s own skills may be the key to improvement. Further research should focus on developing specialized educational courses and decision-support tools for primary care physicians and examining the impact of interprofessional pain management teams on patient outcomes.

## 1. Introduction

The perception of pain is a major public health problem [1]. The basic premise of its treatment is to provide comprehensive support to patients experiencing pain so that they can re-engage in daily life and their previously performed duties [2]. Most often, the responsibility for the treatment of pain and the initiation of therapy with opioid drugs lies with primary care physicians. Patients’ easy access to general practitioners positions them as a key link in pain management. This can lead to many challenges and potential problems due to a lack of education and experience [3]. Deficiencies in physician education are observed both during medical school education and during internships in hospitals and apply to the rules governing opioid use, mastering the management of side effects, and knowledge of opioid abuse and misuse [4,5,6,7]. These deficiencies, when not supplemented with adequate training, become significant knowledge gaps.

Reviewing the existing literature, we noted the multifaceted nature of physicians’ educational needs regarding the proper use of opioid medications [7,8]. Opioids are a class of potent pain-relieving drugs. Opioids are highly effective and safe analgesics, and their appropriate use by competent clinicians is an important element of modern pain management. Critical side effects include respiratory depression, drowsiness, constipation, and nausea. Long-term use can lead to physical dependence and withdrawal symptoms [3]. Patients undergoing opioid treatment should remain under the close supervision of well-educated medical professionals, who may be well-trained family medicine physicians. If the treatment used is not successful, the strategy should be thoroughly reviewed and adjusted [9]. The correct dose of a given opioid is the lowest possible dose that provides the desired effect. When it is necessary to use the highest doses of an opioid, if the pain is still too intense or side effects are maximized, it is advisable to try to implement an alternative opioid drug [3,9,10]. The conversion of opioid doses is not based only on a calculation but should also take into account many other factors (health condition, age, treatment tolerance, and undesirable effects) [11].

In Poland, recommendations in accordance with the World Health Organization’s analgesic ladder are followed. They specify treatment according to the severity of pain, ranging from the use of OTC painkillers (for mild pain) to opioid drugs (for moderate-to-severe pain) [12,13,14,15,16]. However, the international literature repeatedly indicates that most often, opioids are not prescribed because of concerns about a patient’s addiction or abuse of the drug [3,17,18]. Moreover, doctors’ decisions are also influenced by other issues, such as previous professional experience in opioid prescribing [17,18], concerns about professional competence and level of education in pain management and opioid drug use [18,19,20], patients’ reluctance to take opioid drugs [21], concerns about the impact of opioid drugs on patient behavior [3,22,23], the degree of faith in opioids as an effective pain management option [18,20,22], concerns about comorbidities and potential adverse events that could result in suboptimal pain management [6,18,24], complicated procedures for prescribing opioid drugs [22,23], a lack of knowledge of standardized guidelines for opioid use in Europe [3,25] and insufficient time and available resources [26].

Studies indicate that physicians’ greater belief in the use of opioid drugs for cancer pain compared to non-cancer pain is a common phenomenon [6,27]. The majority of family physicians (83%) consider opioids to be an effective treatment for non-cancer pain. At the same time, they worry about the long-term use of these drugs and the management of their doses, which seems to be a very significant inconvenience for them in their daily clinical practice [22]. It was found that the most “liberal” approach to the topic of pain management with opioids is correlated with young physician age and having experience working in specialized oncology/palliative care units [28,29]. Many physicians find the care of patients requiring opioid medications stressful [30]. Analyses conducted in this field have shown that a doctor with extensive experience and a significant number of patients taking opioids under their care has greater confidence and a higher sense of comfort [18,31]. In contrast, a negative belief based on the finding that many patients quickly become addicted to opioids was correlated with the much less frequent prescription of such drugs [18,31]. Due to the human right to live without pain, it is necessary to understand the educational needs of primary care physicians and implement appropriate interventions to improve their knowledge. Therefore, this research study aims to investigate the educational needs of primary care physicians regarding opioid therapy and pain management in the Polish health system setting.

## 2. Materials and Methods

### 2.1. Study Design and Settings

An anonymous computer-assisted web interview (CAWI) survey was prepared as an online questionnaire directed toward doctors working in primary health care (PHC). The study was voluntary, and informed consent was collected from participants. The Checklist for Reporting Results of Internet E-Surveys (CHERRIES) was used [32]. The survey was divided into a section collecting demographic data (5 questions) and employment information and a content section (9 questions).

The survey was developed after performing an intensive literature review on the paper’s topic and a series of interviews between the authors and key individuals related to the scope of the study. The interview answers were collected in the form of a questionnaire, using questions with open-answer options as well as Likert-scale questions. There was also a “non-response” option.

The consent of the Bioethical Committee of the Medical University of Wrocław, decision number KB 472/2020, was obtained.

For the survey, the following combination of tools was used: an online data collection platform, typeform.com, a Google Analytics web analytics system and an .xls file data collection system, that is, Google Docs. This verified that each user filled out the survey only once (IP) and also examined the level of survey abandonment during completion. These tools were launched and set. A pilot study was then performed on a group of 10 doctors, checking the clarity of the questions and how they were answered. Minor modifications were made to better clarify the issues the survey investigated.

The methodology for measuring opioid use and knowledge was based on answering 9 single-choice questions related to these issues. Key from the perspective of the problem at hand were questions addressing the following:Criteria for deciding which painkillers to use;Knowledge related to morphine substitutes and their proper doses;Knowledge related to the management of specific patients;Opioids with the highest prescription rate.

Statistical significance for the studied population of physicians was achieved.

### 2.2. Data Assembly (Study Period and How It Was Performed) and Collection

The questionnaire was made available to respondents in the following time frame: from 10 June 2020 to 10 September 2021. The timing of the research data collection coincided with the SARS-CoV-2 virus pandemic. The authors decided to carry out the survey in the only manner that provided full security to the respondents, i.e., through an online questionnaire. Information about the survey was communicated through classical channels (oral and written invitations; information disseminated at medical events) and close online channels (a newsletter, a website and a forum for primary care physicians in a group on facebook.com) which aimed to invite all doctors to the survey, regardless of their daily use of electronic media. The survey was available on the websites of the Polish Society of Family Medicine and the Polish Society of Palliative Medicine. The survey was open; it was not mandatory for any visitor who wanted to visit the website to complete it, and no incentive was offered in exchange for completing the survey. The criterion for inclusion in the study was working in primary health care. Participants declaring that they did not work in primary care and those who did not provide consent to participate were excluded from this study. The studied population is indicated in Figure 1.

The research data were made available on the Polish Society of Family Medicine website (a trusted third-party site).

### 2.3. Statistical Approach

For the studied group of doctors working in primary care, the minimum number of respondents at the confidence level α = 0.95, at a fraction size of 0.5 and with a maximum error of 5% was 382 respondents. At the same time, it should be stated that for the subgroup of doctors specializing in family medicine, the number of respondents should have exceeded 372, and for doctors with work experience longer than 1 year, the number of respondents should have exceeded 380.

Statistica 13.1 was used for a statistical analysis. In the statistical analysis, the Shapiro–Wilk test was used to test normality. Furthermore, medians and quartile values (Q25, Q75) were used. Utilizing the chi-square test with a significance threshold of *p* < 0.05, this study assessed the occurrence of statistically significant differences within subpopulations of the examined physicians. This assessment included physicians with and without specializations in family medicine. Further, the assessment compared those working up to 40 h per week versus those working over 40 h. Finally, all participants were categorized into subgroups based on the duration of their professional experience. The findings are detailed in tables that are included in the Appendix A of this publication.

## 3. Results

### 3.1. General Characteristics of Participants

Of all respondents, more than 85% had a professional length of service as a primary care physician of at least one year, with more than half of the doctors working in PHC for at least six years. The vast majority of respondents mentioned PHC as their main place of work (87.6%), and only a minority of them (3.3%) worked in palliative care at the same time. Over 77% (Q75) of the respondents to the questionnaire were family medicine specialists or in training for this specialization. The occupational characteristics of the respondents are presented in Table 1.

### 3.2. The Choice of Pain Medication and Opioid Prescription

The first question of the survey concerned the main criterion that determined the choice of pain medication (Table 2). Family physicians in Poland claim that they choose analgesics in accordance with current medical knowledge, based on a clear clinical rationale, such as pain intensity, as measured using a pain rating scale (53% of respondents), the nature of pain (27%) and the rate of pain changes (12%) (Table 2). Another important issue that respondents were asked about was the writing of prescriptions for opioid drugs. Nearly 54% of general practitioners said they initiate pain treatment with weak or strong opioids as needed. There was no statistical significance of the observations in the subgroup of examined physicians [Appendix A].

### 3.3. Primary Care Physician Knowledge on the Topic of Opioid Usage

The respondents were then asked two separate questions about morphine and its substitutes, that is, to indicate what doses of morphine correspond to 400 mg of tramadol and 35 μg of buprenorphine (Table 3). The question was formulated based on the knowledge requirements from the specialization examination for family doctors in Poland. The purpose of this question was to objectify doctors’ declarations regarding the use of painkillers. As many as 42% and 48% of general practitioners, respectively, declared no knowledge or skills to convert the correct dose of morphine. Moreover, a further 39% (for tramadol conversion) and 33% (buprenorphine) incorrectly converted doses. In the case of a patient with NRS 6/10 pain, only every fourth respondent indicated the need to use drugs from the third step of the analgesic ladder.

Subsequently, the study participants were presented with two clinical cases for which they were asked to suggest a course of action to reduce patients’ pain. Table 3 shows the numerical values corresponding to the number of specific responses. Almost 73% of doctors do not attempt to adjust the dose of morphine despite the presence of obvious clinical indications.

Another question in the survey asked about the opioid medications most commonly used in daily medical practice (Table 4). The analysis shows that only 3.7% of respondents use an orally administered morphine treatment. Statistical significance was not achieved for the observation of declarations of preferable opioid drugs chosen by physicians.

Further analysis showed that nearly 72% of respondents choose to implement coanalgesic drugs when pain of significant severity is diagnosed (Table 4). One of the most important questions for the problem at hand, also concerned the identification of the most difficult element of pain treatment in the daily practice of primary care physicians (Table 4). No statistical significance was demonstrated between study subgroups [Appendix A].

### 3.4. Comparison of the Right Clinical Decision with the Percentage of Doctors Declaring Opioid Use

In the surveyed group of physicians, a declarative 84% include treatment with drugs from levels II and/or III of the analgesic ladder (Figure 2). Furthermore, as little as 25% of physicians initiating opioid therapy and less than 10% of physicians using tramadol or buprenorphine in their practice correctly recalculate opioid doses, while when asked about the practical use of opioid drugs, 64% and 27%, respectively, incorrectly select clinical management. The group of examined doctors with specialties other than family medicine in the entire study group did not have a statistically significant impact on the overall result, which was calculated.

## 4. Discussion

### 4.1. Main Findings

The main findings of this study show the disproportion between the declarative use of opioids in pain therapy (84% of respondents) and actual knowledge regarding their administration (the correct conversion of opioid doses in the case of 9% of physicians ordering only tramadol or buprenorphine and 25% of physicians initiating opioids).

Previous studies were conducted mainly on groups other than primary care physicians (pre-graduation), as in the study by Pieters et al. [4]. In the systematic review of Hooten et al., physicians reported a high level of awareness of the potential for opioid misuse and were concerned about inadequate prior training in pain management [6]. In the opinion of the authors, based on the collected results, this fact can be questioned. Doctors overestimate their skills and incorrectly define their state of knowledge.

In a 2014 study by Jamison et al. examining primary care physicians, respondents expressed concern about medication misuse (89%) and felt that managing patients with chronic pain was stressful (84%). Most were worried about addiction (82%). At the same time, only less than half felt that they were sufficiently trained in prescribing opioids (46%) [33]. An analysis of the scientific literature indicates that physicians have a limited ability to reliably self-assess their skills and competencies [34,35,36,37]. Significant discrepancies between the belief in one’s own knowledge about the use of opioid analgesics and one’s actual skills in this area carry significant risks for patients suffering from pain [34]. A primary care physician convinced of their own skills is at risk of making more mistakes, mistaking a drug or its dosage and finally, may have difficulty meeting the basic tenets of effective pain therapy (adequacy, avoiding side effects) [34].

The authors agree with the general conclusions of the systematic review by Carey et al., which indicated available comprehensive training for primary care practitioners is likely needed to address issues of low confidence [37]. In our opinion, it is important not to base the development of training solely on the subjective needs reported by primary care physicians. There should not only be dedicated training at the pre-graduation stage. They must (not only “ideally” [37] but actually) take into account the needs of primary care physicians, including the inadequate assessment of skills in relation to their actual usage in clinical practice. In their work, Green at al. draw attention to the five key roles of the primary care physician in palliative care from the points of view of patients and their caregivers. They include knowledge and competence. The perspective of the patient, who is predominantly unable to assess the correctness of the doctor’s work, is important in our opinion [38]. Difficulty arises when, as we have shown, the doctor does not correctly assess their skills.

### 4.2. What This Study Provides

The factor in the form of a subjective belief in one’s own skills that stands in contrast to actual skills should, in our opinion, be added to the deficiencies in education, accessibility difficulties, prejudice and legislation identified in the literature as another factor that significantly limits the proper management of pain treatment. An important element to consider, therefore, when designing educational and clinical decision support solutions and targeting primary care physicians should be the proper determination of actual, rather than declarative, knowledge [33].

Juxtaposing data obtained from questions about actual clinical situations with data on the use of specific drug groups helps identify issues that need to be addressed in educational programs for primary care physicians. Properly conducted research can also help create an educational program for training doctors. Doctors claim difficulties in titrating morphine, for example. However, they mention it as the least frequently used opioid drug. The declaration (in terms of the frequency of morphine use) juxtaposed with the description of the difficulties (lack of titration skills) indicates the most important issues that need to be addressed when creating educational programs for doctors.

The authors of this paper carried out a process of creating an educational course and a decision support solution for opioid medication rotation [9] based on the results of the presented study.

One of the important elements for improving the effectiveness of the implemented pain treatment strategies and improving the safety of the applied therapy in the primary care setting is the provision of access to pain management teams [37,39] through the presence of interdisciplinary teams giving the family doctor the opportunity to consult or by referring the patient to readily available pain management teams (clinics).

### 4.3. Strengths and Limitations

The survey covered a significant number of primary care physicians working in Poland, ensuring a broad representation of this group and increasing the generalizability of the results, while the use of an online survey for data collection allowed for easy access, reduced the data collection time and acquired responses from physicians across the country. The response rate is consistent with that expected from the literature [40] and from similar studies conducted in Poland using the CAWI method. The study focused on determining the educational needs of primary care physicians in terms of not only declarative knowledge or skills possessed but both.

The results of the survey are based on self-reported assessments of physicians regarding their knowledge and skills, which may not fully reflect their actual behavior in clinical practice, though it does allow for an assessment of the discrepancy between self-assessments of their competence and their actual skills. The research design did not include any interventions, which limits the ability to establish causal relationships between educational programs and improved pain management. Although the authors made efforts to obtain representativeness in the group, statistical significance was obtained, and a group of over 1% of all primary care physicians was examined; it should be noted that in this study, no selection was made of the surveyed participants in terms of representation (gender, age, or place). It should be noted that for doctors specializing in family medicine for *p* < 0.05, the required number of study participants was not achieved. Subsequent research should focus on this group to determine whether the observed differences are statistically significant.

The aim of this study was not to determine the circumstances of introducing opioid drugs in primary health care. Further research should focus on identifying whether primary care physicians effectively implement and modify treatment depending on the nature of the pain (acute pain; chronic pain). Future research studies can focus on types of problems that doctors encounter in patients treated chronically with opioids and their correlation with treatment errors. This may provide an answer as to whether the side effects that patients complain about are correlated with the properties of a drug or its improper use.

This study showed a significant number of incorrect answers and a lack of self-awareness of doctors when providing them. Further research is required to determine the causes of these errors, the correlation between knowledge and treatment management and how to address this via education programs or other solutions.

## 5. Conclusions

The proper self-assessment of one’s own skills may be the key to improving pain management in primary care. Targeted educational programs are essential to bridging the knowledge gap and increasing physicians’ competence in opioid therapy and pain management. Solutions that objectify pain treatment should be implemented in everyday medical practice (both the use of appropriate tools and scales and support for clinical decision making). Further research should focus on developing specialized educational courses and decision-support tools for primary care physicians and examining the impact of interprofessional pain management teams on patient outcomes.

## Figures and Tables

**Figure 1 healthcare-12-00217-f001:**
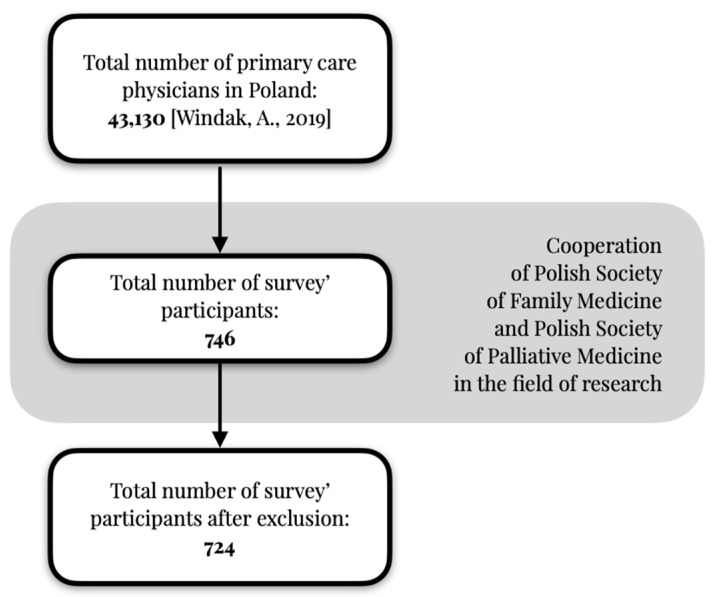
Studied population of primary care physicians [32].

**Figure 2 healthcare-12-00217-f002:**
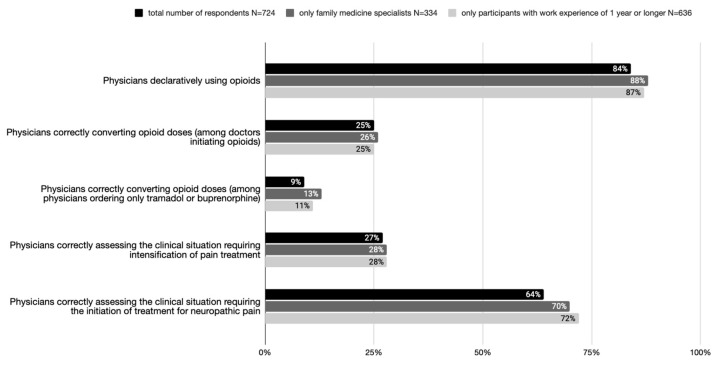
Percentage of doctors making the right clinical decision compared to percentage of doctors declaring opioid use.

**Table 1 healthcare-12-00217-t001:** Participant characteristics.

Participant Characteristics	N (%)
Professional length of service as a primary care physician (n = 744)	
Less than 1 year	92 (12.4)
From 1 year to 5 years	244 (32.8)
From 6 years to 15 years	182 (24.5)
More than 16 years	206 (27.7)
I do not work now in primary health care/not a doctor	20 (2.7)
Primary health care is the main place of work (n = 724)	
Yes	634 (87.6)
YES, and I simultaneously work in palliative care	24 (3.3)
No	54 (7.5)
NO, and I simultaneously work in palliative care	12 (1.7)
Average number of hours spent in primary care per week (n = 724)	
Less than 10 h per week	32 (4.4)
10–20 h per week	42 (5.8)
20–40 h per week	464 (64.1)
More than 40 h per week	186 (25.7)
Respondents’ medical specialties (n = 724)	
Family medicine specialist	334 (46.1)
Undergoing specialization in family medicine	226 (31.2)
Internal medicine specialist	124 (17.1)
Palliative medicine specialist	12 (1.7)
Pediatric specialist	20 (2.8)
Other	18 (2.5)
None	88 (12.2)

**Table 2 healthcare-12-00217-t002:** Criteria for the choice of pain medication and opioid prescription statements.

Question	Answers	Total Group, N (%)	Only Family Medicine Specialists, N (%)	Only Participants with Work Experience of 1 Year or Longer, N (%)
		724 (100)	334 (100)	636 (100)
Criterion for determining the choice of pain medication, according to respondents.	Pain intensity measured by pain rating scale	387 (53.5)	172 (51.5)	332 (52.2)
	Nature of pain	201 (27.8)	96 (28.7)	174 (27.4)
	Dynamics of pain evolution; speed of escalation	88 (12.2)	40 (12.0)	78 (12.3)
	Own experience with a particular drug	44 (6.1)	18 (5.4)	44 (6.9)
	Location of pain	4 (0.1)	2 (0.6)	3 (0.5)
Do you write prescriptions for opioid drugs?	I initiate pain management with weak or strong opioids as needed	388 (53.6)	182 (54.4)	354 (55.6)
	I use tramadol and buprenorphine preparations (in patches)	234 (32.3)	112 (33.5)	200 (31.4)
	I write opioid drugs, but only as a continuation of therapy	92 (12.7)	26 (7.8)	64 (10.6)
	I use only tramadol	29 (4.0)	12 (3.6)	16 (2.5)

**Table 3 healthcare-12-00217-t003:** Practical questions testing the ability to calculate the dose of an opioid drug and case studies checking the ability of primary care physicians to properly manage pain treatment.

Question	Answers	Total Group, N (%)	Only Family Medicine Specialists, N (%)	Only Participants with Work Experience of 1 Year or Longer, N (%)
		724 (100)	334 (100)	636 (100)
What dose of morphine is equivalent to 400 mg of tramadol (approximate, in accordance with Polish guidelines)?	I don’t know	307 (42.4)	136 (40.7)	258 (40.6)
	40 mg	152 (21.0)	66 (19.8)	128 (20.1)
	**80 mg**	**133 (18.4)**	**70 (21.0)**	**114 (18.0)**
	20 mg	100 (13.9)	38 (11.4)	88 (13.9)
	60 mg	51 (7.0)	22 (6.6)	45 (7.1)
What dose of oral morphine is equivalent to 35 mg of buprenorphine (tts, approximate, in accordance with Polish guidelines)?	I don’t know	350 (48.3)	142 (42.6)	280 (44.0)
	**80 mg**	**135 (18.6)**	**72 (21.6)**	**124 (19.5)**
	20 mg	133 (18.4)	64 (19.2)	118 (18.6)
	40 mg	68 (9.4)	26 (7.8)	62 (9.8)
	60 mg	57 (7.9)	28 (8.4)	50 (7.9)
The patient takes 400 mg (tramadol retard per day in two divided doses. In addition, he takes paracetamol 500 mg every 8 h. Current pain severity is 6/10 on the NRS scale. What treatment will you suggest (in accordance with Polish guidelines)?	Initiate treatment with buprenorphine patch, starting with a dose of 35 μg/h (tts) and gradually increase the dose every 6 days or so as needed until control is achieved	254 (35.1)	128 (38.3)	228 (35.9)
	**Initiate treatment with short-acting oral morphine at a dose of ½ tablet of 20 mg every 4 h + emergency analgesic dose and titrate until control is achieved**	**196 (27.1)**	**94 (28.1)**	**180 (28.3)**
	A coanalgesic, such as pregabalin, should be included first	164 (22.7)	58 (17.4)	124 (19.5)
	Replace paracetamol with ketoprofen and give it twice daily at a dose of 100 mg	39 (5.4)	22 (6.6)	36 (5.6)
	Increase the total dose of tramadol by 1/2 to a total of 600 mg in two divided doses	29 (4.0)	8 (2.4)	22 (3.5)
	None (no treatment)	62 (8.6)	22 (6.6)	44 (6.9)
The patient, 57 years old, was treated about 3 months ago for hemiplegia of the facial area on the left side. Since then she is constantly accompanied by pain in this area 4/10 on the NRS scale treated with oral tramadol 2 × 100 mg, hypersensitivity and burning sensation. What treatment would you suggest (in accordance with Polish guidelines)?	**You will join the treatment with pregabalin starting with a dose of 75 mg twice a day**	**542 (74.7)**	**232 (69.5)**	**456 (71.2)**
	You will join the treatment with gabapentin trying to reach a dose of 3 × 300 mg	160 (22.1)	80 (24.0)	146 (23.0)
	You will include a drug from step 3 of the analgesic ladder in the face of tramadol’s ineffectiveness	21 (2.9)	8 (2.4)	8 (1.3)
	You will repeat treatment with acyclovir at a dose of 4 × 400 mg	4 (0.6)	8 (2.4)	14 (2.2)
	None (no treatment)	17 (2.3)	4 (1.2)	10 (1.6)

Abbreviations: mg—milligrams; μg/h—micrograms per hour; NRS—Numeric Pain Rating Scale. Correct answers are shown in bold.

**Table 4 healthcare-12-00217-t004:** Most chosen opioids (declarations), coanalgesic drug introduction and the most difficult part of pain treatment in daily primary care practice.

Question	Answers	Total Group, N (%)	Only Family Medicine Specialists, N (%)	Only Participants with Work Experience of 1 Year or Longer, N (%)
		724 (100)	334 (100)	636 (100)
Most commonly used opioids in daily medical practice	Tramadol (oral)	490 (67.7)	192 (57.5)	404 (63.6)
	Buprenorphine in a patch	170 (23.4)	94 (28.1)	160 (25.2)
	Oxycodone (oral)	34 (4.7)	26 (7.8)	32 (5.0)
	Short-acting oral morphine	27 (3.7)	14 (4.2)	20 (3.1)
	I do not use opioids in my medical practice	24 (3.3)	4 (1.2)	12 (1.9)
The stage of treatment at which coanalgetic drugs are introduced	At the time of diagnosis, that the pain is neuropathic in nature	518 (71.5)	226 (67.7)	448 (70.4)
	When drugs from step 3 of the analgesic ladder are ineffective	129 (17.8)	60 (18.0)	108 (17.0)
	At the first signals from the patient regarding pain	45 (6.2)	24 (7.2)	38 (7.1)
	Other	14 (1.9)	4 (1.2)	8 (1.6)
	I never include coanalgesics in my practice	37 (5.1)	16 (4.8)	24 (3.8)
The most difficult part of pain management in daily PCP practice	Morphine titration in a primary health care/home treatment	322 (43.4)	138 (41.3)	276 (43.4)
	Replacing one painkiller with another (rotation)	150 (20.7)	76 (22.8)	128 (21.1)
	Controlling breakthrough pain	98 (13.5)	46 (13.8)	86 (13.5)
	Adding coanalgesics to pain management	86 (11.9)	38 (11.4)	72 (11.3)
	Selecting non-steroidal anti-inflammatory drugs according to the type of pain	51 (7.0)	16 (48.0)	38 (6.0)
	Other	4 (0.6)	2 (0.6)	4 (0.6)
	None	33 (4.6)	16 (4.8)	30 (4.8)

## Data Availability

The data and materials generated during this study are available freely upon request from the Polish Society of Family Medicine web page.

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
