# Peer review of "Opioid Treatment in Primary Care: Knowledge and Practical Use of Opioid Therapy"

_healthcare, 2024, doi:10.3390/healthcare12020217_

Round 1

Reviewer 1 Report

Comments and Suggestions for Authors

1. The work lacks a short introduction to opioids - what group of drugs it is, what is its classification, what is its use, and what are the most critical side effects.

"The consent of the Bioethical Committee of the Medical University of Wrocław was obtained" - please provide your consent number.

2. please provide the inclusion and exclusion criteria for participants in the study. It is unclear whether the questions were single-choice or multiple-choice.

3. The number of physicians in table 1 (n=744, n=724) does not agree with figure 1 (n=746 or n=728). This is an error or analysis of partial (incomplete) data - a methodological error.

4. the answers should be correlated with the physician's specialization. Table 1 contains a question about working in palliative care. It is known that this is a group of physicians who deal with pain daily, and the use of strong opioids is common there.

Moreover, some respondents do not work in primary care as their main job. The use of opioids by an anesthesiologist, urologist or obstetrician will be completely different, and this is not due to a lack of knowledge or skills, but to specialization in a different field. This aspect may have been lost in the study.

4. Table 1. The 16 people presented in the study are not physicians, which should exclude them from the study and statistical analysis. Again, there are no exclusion criteria and exclusion of inappropriate trials.

5. 92 physicians work less than 1 year, and 88 physicians are without specialization. In my opinion, it should be an exclusion criterion, because it's hard to say about experience if you are working for few months. Opioid addcition, tollerance, sidde efffects, withdrawal are long term problems, thats cannot be properly assessed in such a short time.

6. 3.2. The choice of pain medication and opioid prescription. Lines 158-166. What about "WHO analgesic ladder"? It is a basic framework created by the WHO to guide physicians on appropriate pain relief strategies. It involves a three-step progression from non-opioids to weak opioids to strong opioids until sufficient pain relief is obtained. Around this central framework, adjuvant medications and procedural options are also given and recommended when deemed necessary due to ineffective pain control.

7. Table 2. Why are tramadol and buprenorphine together in one question?

These are drugs with different therapeutic profiles, different routes of administration, different pharmacokinetic profiles, potency, and receptor effects. Opioids cannot be categorized only as a group of drugs with different potency.

Buprenorphine, for example, a partial/reverse agonist, has activity in neuropathic pain, is a very potent agonist of the μ-opioid and nociceptin receptors, and an antagonist of the δ and κ opioid receptors. A characteristic feature of partial agonists of opioid receptors is the so-called ceiling effect, which means that after a certain dose, the analgesic effect of the drug decreases. However, no ceiling effect was observed for buprenorphine in clinical trials despite its use in high doses.

Meanwhile, tramadol, a pure agonist, works by connecting to the μ, d and k opioid receptors. It also affects pain transmission in the spinal cord by increasing the concentration of certain neurotransmitters (serotonin, norepinephrine) in synapses; therefore, it may cause side effects not typical of opioids, such as anxiety, agitation, insomnia, malaise, nausea, and vomiting.

Also in the classification of opioid strength - tramadol is classified as a weak drug and buprenorphine as a strong drug. The question should be extended to the division into weak and strong drugs (safety of use, strength of the drug = side effects, risk of addiction), routes of administration: tablets, patches, drops (the physician's understanding of the principle of action of the drug in acute pain, breaking pain, chronic pain, and atypical pain, quick and strong and long-lasting effect = optimization of the route of administration).

Question: "Do you write prescriptions for opioid drugs?" it also seems poorly constructed. Can pain treatment be limited to only 4 alternatives? The question should be open-ended and then categorized, or it should include a list of all available opioid medications, including NSAIDs (as an alternative to opioids), to allow for an answer that is closer to the truth. This alternative is only found in Table 4, which causes inconsistency between the questions.

Physicians who do not use opioids (table 4) cannot choose the correct answer, then when asked about dose conversion, they give wrong answers - because they do not use these drugs routinely. Frequency of use in practice should be correlated (1:1) with knowledge of dosage and regimens. If a physician does not use a drug, he or she is not making an error in its use. The work should answer the question - do physicians who routinely use opioid drugs (neurologists, surgeons, anesthesiologists, oncologists) use them correctly?

8. Table 3, 4.

The alternatives "all of the above" and "none of the above" should not be used in medical survey. When "all of the above" is used as an answer, test-takers who can identify more than one alternative as correct can select the correct answer even if unsure about other alternative(s). When "none of the above" is used as an alternative, test-takers who can eliminate a single option can thereby eliminate a second option. In either case, user can use partial knowledge to arrive at a correct answer and the option "none of the above" does not test whether the tested person knows the correct answer, but only that he/she knows the distractors aren't correct.

table 4.

Question: The most difficult part of pain management in daily PCP practice. Answer "None of the above" mean the same as "Others". It should be changed to "I have no problem with pain management". This construction of question force the physician to indicate that use of opioids is problematic.

Question: "The stage of treatment at which coanalgetic drugs are introduced" does not gives significant knowledge. The fact that a physician does not administer a co-analgesic does not necessarily indicate a treatment error. You may not have patients with neuropathic pain in your practice. The last question from table 3 gives the correct answer about the physician's competencies.

To sum up:

the authors completely omitted:

- differentiation of treatment their patients: occasionally (sciatica, postoperative pain) from patients treated chronically (cancer pain),

- distinguishing the nature of pain vs. selecting a drug - acute (traumatic) pain, chronic, inflammatory, neuropathic pain. The correlation between drug and co-analgesic selection is as important as dose selection,

- long-term drug effect - what problems do doctors encounter in patients treated chronically with opioids - such knowledge, correlated with treatment errors, may only provide an answer as to whether the side effects that patients complain about (and that doctors are afraid of) are actually correlated with the properties of the drug or its improper use,

- there is no analysis of incorrect answers - what caused them, what kind of the error was, and how can it be fixed in the future?

- again, the lack of correlation makes it impossible to understand which group of physicians makes what kind of error and, on the other hand, which doctors cope well with pain therapy.

The strength of survey work is to look for correlations - between knowledge and the treatment regimen, drug selection and the disease, knowledge of the regimens/rotation and the effectiveness of treatment and the frequency or potency of side effects. And so on.

The work has potential and points to a significant problem of lack of proper knowledge about treatment - in this case, pain. Unfortunately, the structure of the questions causes the collected information to be distorted or incomplete. Researchers make a mistake when preparing questions tailored to their thesis (no answers - I have no problems, I do not use opioids, selective selection of drugs to choose from, only TRA and BUP regimens without the opportunity to demonstrate other knowledge). Meanwhile, the question should be neutral and allow for a broader answer across the entire scale. Next, the authors did not provide any advantages of physicians, their knowledge strengths, what areas of they knowladge and skill are correct, when pain treatment is not questionable. Every coin has two sides - the authors presented only one.

Reviewer 2 Report

Comments and Suggestions for Authors

Article with a very current topic, relevant, but difficult to approach. I believe that your approach will be an added value and an act of courage.

Clear and objective writing that is easy to understand.

Title could be more objective.

Objective summary, with a well-defined objective, explaining the research methodology used. It highlights the results obtained, conclusions and suggestions for future work to be carried out.

Keywords – put 5, adjusted and according to the Mesh descriptors.

Regarding the bibliographical review, I think they focus on an interesting topic, which for many is impossible to happen. Clinicians' lack of knowledge about pain and its opioid treatment will have direct implications for people's well-being, as they are the possible prescribers of this type of therapy. They address the topic of pain and its control, focusing on the difficulties and constraints of prescribing clinicians, namely the lack of approach to this topic in the theoretical and practical context of the course.

They clearly define the objective of this study.

Methodology – define the methodology. They identify the questionnaire and refer to its constitution and the areas covered.

Clearly identifies the period for data collection, online during the pandemic period as it is the most advantageous. They could have been clearer regarding its constraints and the justification for the exclusion of the 18 participants, as shown in figure 1.

The research was open to all interested parties and was publicized online and through medical events.

They give an idea of ​​the population for which the research is intended, and the sample obtained, characterizing the elements that constitute it.

Regarding the results, these are worrying. It was identified that many clinicians do not have the knowledge that allows them to act correctly in ​​pain control. In this way, they opt for attitudes that are not the most appropriate.

The authors propose the creation of a training course in this area and highlight the importance of pain management teams.

The limitations identified are relevant. They do not clearly identify suggestions for future work and do not give a very objective perspective of what they intend to do in terms of educational programs aimed at this specific population in the context of increasing knowledge about pain and its correct treatment. The fact that 12.4% of participants had less than 1 year of service may also be significant. It would be important to make some relationships between variables to obtain more consistent results.

In conclusion, they identify the pressing need to invest in training programs in this area, which are developed with this specific population so that better and more effective results can be obtained in the control and treatment of pain, as a right of all human beings.

As for the bibliography and given that it is such a current and studied topic, although not specifically in this sample, I consider that some dates are very old (1986, 1999) and could be more current, especially in terms of theoretical foundation. The 1999 reference relates to the epidemiology of chronic pain in the community, being a very important topic and not justifying such an old date. They present 40 bibliographic references, 7 of which are from 2019 and 2 from 2021 and 2018, 1 from 2023. The rest are older, more than 10 years old, which does not seem to be very appropriate, given the current nature of the topic.

I suggest readjusting the bibliography; greater commitment to identifying exclusion criteria and greater objectivity in suggestions for future work.

Round 2

Reviewer 1 Report

Comments and Suggestions for Authors

The work has been corrected and is suitable for publication.

In the future, please answer the reviewer's questions below, it will make it easier to find the problem the question was about and leave the line numbers in rev2.